# Reproducibility in Multiple Instance Learning: A Case For Algorithmic Unit Tests

**Edward Raff**
Booz Allen Hamilton
University of Maryland, Baltimore County
`raff_edward@bah.com`

**Jim Holt**
Laboratory for Physical Sciences
`holt@lps.umd.edu`

## Abstract

Multiple Instance Learning (MIL) is a sub-domain of classification problems with positive and negative labels and a "bag" of inputs, where the label is positive if and only if a positive element is contained within the bag, and otherwise is negative. Training in this context requires associating the bag-wide label to instance-level information, and implicitly contains a causal assumption and asymmetry to the task (i.e., you can't swap the labels without changing the semantics). MIL problems occur in healthcare (one malignant cell indicates cancer), cyber security (one malicious executable makes an infected computer), and many other tasks. In this work, we examine five of the most prominent deep-MIL models and find that none of them respects the standard MIL assumption. They are able to learn anti-correlated instances, i.e., defaulting to "positive" labels until seeing a negative counter-example, which should not be possible for a correct MIL model. We suspect that enhancements and other works derived from these models will share the same issue. In any context in which these models are being used, this creates the potential for learning incorrect models, which creates risk of operational failure. We identify and demonstrate this problem via a proposed "algorithmic unit test", where we create synthetic datasets that can be solved by a MIL respecting model, and which clearly reveal learning that violates MIL assumptions. The five evaluated methods each fail one or more of these tests. This provides a model-agnostic way to identify violations of modeling assumptions, which we hope will be useful for future development and evaluation of MIL models.

## 1 Introduction

In Multiple Instance Learning (MIL) we have a dataset of $N$ labeled points, which we will represent as $\mathcal{X}$ with associated labels $y \in \{-1, 1\}$ for the negative and positive labels respectively. As originally described, the MIL problem involves each datum $X_i \in \mathcal{X}$ being a *bag* of multiple *instances*, where $X_i = \{\mathbf{x}_1, \mathbf{x}_2, \ldots, \mathbf{x}_{n_i}\}$ is a bag of $n_i$ instances. Each instance $\mathbf{x}_j \in X_i$ is a $D$-dimensional vector, and every bag $X_i$ may have a different total number of items $n_i$. Given instant level classifier $h(\cdot)$, most MIL algorithms work by predicting $\hat{y}_i = \max_{\forall x_j \in X_i} h(x_j)$.

As originally described, the positive/negative label of each bag $X_i$ has a special meaning. By default, a bag's label is negative ($y = -1$). The label of a bag will become positive ($y = 1$) if and only if a positive instance $\mathbf{x_j}$ is present inside the bag, at which point the entire bag's label becomes positive. Because instance-level labels mapping each $\mathbf{x_j} \rightarrow y \in \{-1, 1\}$ are not given, the MIL problem is to infer the instance-level labels from the whole-bag level labeling. This implies a critical asymmetric nature to the given labels and how they must be handled. A value of $y = -1$ tells us that all instances are negative in the given bag, whereas a label of $y = 1$ tells us that one or more instances have a positive label. For this reason, swapping the positive and negative labels in a MIL problem is not

37th Conference on Neural Information Processing Systems (NeurIPS 2023).

semantically meaningful or correct, whereas, in a standard classification problem, the labels can be interchanged without altering the semantics of the learning task.

The MIL problem occurs with frequency in many real-world applications, in particular in the medical community where the presence of any abnormal cell type (i.e., instance) is the confirmative indicator for a larger organism's disease (i.e, bag and label). As the MIL problem implies, and the medical example makes explicit, the MIL model has an implicit casual assumption: the right combination of positive indicators dictate the output label, and so the MIL model is both a valuable inductive bias toward the solution and a guard against physically implausible solutions.

Algorithms that fail, or intentionally forgo, the MIL constraints may appear to obtain better accuracy "in situ" (i.e., the lab environment). But if it is known that the MIL assumption is true, ignoring it creates a significant risk of failure to generalize "in vivo" (i.e., in real production environments). In the clinical context, this is important as many ML algorithms are often proposed with superior in situ performance relative to physicians [1], but fail to maintain that performance when applied to new clinical populations [2–4]. In this case, respecting underlying MIL properties eliminates one major axis of bias between situ and vivo settings and higher confidence in potential utility. In the cyber security space, respecting the MIL nature eliminates a class of "good word" style attacks [5–7] where inconsequential content is added to evade detection, an attack that has worked on production anti-virus software [8–10]. These reasons are precisely why MIL has become increasingly popular, and the importance of ensuring the constraints are satisfied.

Notably, this creates a dearth of options when more complex MIL hypotheses are required, as CausalMIL and mi-Net succeed by restricting themselves to the Standard MIL assumption. The creation of MIL models that satisfy this, and other more complex hypotheses, are thus an open line of research that would have potentially significant clinical relevance. Similarly, users with more niche MIL needs may desire to more thoroughly test their models respect the constraints critical to their deployment. Our work has demonstrated that many articles have not properly vetted the more basic MIL setting, and so we suspect other more complex MIL problems are equally at risk.

Our work contributes the identification of this issue, as well as a strategy to avoid repeat occurrences, by developing *algorithmic unit tests* where a synthetic dataset is created that captures specific properties about the desired solution. The test will fail if an invariant of the algorithm's solution is not maintained, as summarized in Table 1. Such failures indicate that a MIL method is not properly constrained, and the learning goal is not being achieved. We construct three such datasets for the MIL problem, which can be reused by any subsequent MIL research to mitigate this problem. Based on these results, we would suggest practitioners/researchers begin with CausalMIL and mi-Net as a solid foundation to ensure they are actually satisfying the MIL hypothesis, and thus avoiding excess risk in deployment.

Table 1: Our work proposes a test of the Standard Multiple Instance Learning (MIL) hypothesis and two tests for the Threshold MIL. Most modern deep learning MIL models do not test or prove that they respect any MIL hypothesis, and our tests show that most are insufficient. When a model passes the Threshold test, but fails the Standard test, it is still not a valid Threshold MIL model because the tests do not guarantee correctness, they are like software unit tests that identify failures.

| Model | mi-Net | MI-Net | MIL-Pooling | Tran-MIL | GCN-MIL | CausalMIL | Hopfield |
| Claim | Standard | Standard | Standard | Threshold | Threshold | Standard | Threshold |
| --- | --- | --- | --- | --- | --- | --- | --- |
| Standard Test | ✓ | ✗ | ✗ | ✗ | ✗ | ✓ | ✗ |
| Threshold Tests | ✗ | ✓ | ✗ | ✗ | ✗ | ✓ | ✓ |

This paper is organized as follows. In § 2 we will review broadly related works, including prior work in non-deep-MIL and deep-MIL literature. It is in this related work we will denote the baseline algorithms we test, in particular the five deep-MIL models that form the foundation of most current deep-MIL research, and a sixth deep-MIL method that is little-known but does pass our tests. Next in § 3 we will define three algorithmic unit tests for MIL models. The first tests the fundamental MIL assumption that all models must respect, and the second and third tests extend to a generalized version of the MIL problem known as "threshold" MIL. Prior deep-MIL works might tacitly assume they can tackle the generalized MIL, but make no formal specification of the types of MIL models they tackle. Then we apply our tests to six deep-MIL and seven older Support Vector Machine based MIL models in § 4, demonstrating how different algorithms pass and fail different unit tests. In doing

so we provide hard evidence that the foundations of most current deep-MIL works are invalid, and thus dangerous to use in any case where the MIL assumption is used for casual or clinically relevant constraints. For example, although a cancer diagnosis should occur only because cancer was detected, a non-MIL model could learn the absence of something unrelated as a false signal, causing it to overfit. In addition, we discuss cases where a known non-MIL algorithm still passes the unit test, prompting a discussion on how unit tests should be used for invalidation, not certification. Finally, we conclude in § 5.

## 2  Related Work

Concerns about reproducibility within the fields of machine and deep learning have increased in recent years. Prior works have studied reproducibility issues with respect to dataset labels/integrity [11–13], comparison methodology, and making conclusions on improvement [14–18], discrepancies between math and floating-point precision [19], discrepancies between code and paper, [20], and false conclusions in repeatability [21]. We note that none of these prior efforts on reproducibility would have identified the MIL assumption violation that we identify in this work. Our situation is a different aspect of reproducibility in that the methods under test can be reproduced/replicated, but the methods themselves fundamentally are not designed to enforce the modeling assumptions, and no testing was done to ensure that they do. By developing tests meant to elicit certain and specific behaviors of a MIL model, we show how unit tests may be developed for an algorithm in the form of synthetic datasets.

Within the reproducibility literature, we believe the work by Ahn et al. [19] is the most similar to ours, where they develop a mathematical framework for characterizing the reproducibility of an optimization procedure when initialization and gradient computations are not exact. This is motivated by the fact that proofs of optimization do not account for floating point precision in the majority of cases, so specialized domain tests can be useful. A key point is that having source code is helpful, but does not confer correctness of the property of interest [22]. In contrast, our work is more empirical in that we actually implement our proposed tests, and our tests are born not from a mismatch between math and implementation but from the observation that prior works have neglected the mathematical work to ensure their methods follow the MIL assumptions. Other relevant works in reproducibility have looked at methodological errors [15, 23, 24].

The MIL problem bears resemblance to a niche set of defenses used within the malware detection literature. To defend against adversarial attacks, "non-negative" [5] or "monotonic" [6] models were developed where features can only be positive (read, malicious) indicators, and by default, all files would be marked negative (benign) absent any features. This is similar to the MIL model assumption that there is no positive response unless a specific instance is present, and indeed, MIL approaches have been used to build malware detectors that are interpretable and not susceptible to attack [8].

### 2.1  Relevant Multi-Instance Learning Work

An explosion of interest in MIL literature has occurred due to its relevance in medical imaging and other tasks where the MIL assumption aligns with clinically relevant or important physical/causal constraints of the underlying system. Shockingly, we find much of the literature does not test or ultimately respect this core MIL assumption, resulting in models that are at risk of over-fitting their training data and learning clinically/physically invalid solutions. Simulated benchmarks are common in MIL literature, but focus primarily on the Standard formulation and accuracy [25]. [26] built synthetic benchmarks of MIL tasks, but did not formalize what kinds of MIL tasks or attempt to check if a model was violating the underlying generative MIL hypothesis[1]. The key difference of our work is to create synthetic datasets to test that a model respects the MIL assumptions, rather than benchmark accuracy. We will first review the older, predominantly Support Vector Machine (SVM) history of MIL models that we will test in this work. Then we consider the more recent deep learning counterparts.

---

[1]Two tasks are Standard MIL, one is Threshold MIL, and a fourth is indeterminate but closest to the Generalized MIL of [27]

### 2.1.1 Historical Non-Deep MIL

Issues with under-specification of the MIL problem had been previously identified in the seminal survey of Foulds and Frank [27], who synthesized many implicit MIL extensions into a set of generalized and well-specified MIL types. As noted by this prior work, *many MIL papers from this time period do not provide proof or attempt to enforce the MIL assumption.* Thus while they may be exploring a broader scope of the MIL hypothesis space, the developed solutions may still fundamentally not satisfy the definition of any MIL model.

We will briefly review some significant non-deep MIL models that we include as comparison points, and their status with respect to the MIL assumption. Most notably the mi-SVM and MI-SVM algorithms [28] are correct by construction to the standard MIL model that we will discuss further in § 3.1. The MI-SVM in particular introduces the idea of a "witness", where the bag label is inferred from a singular maximum-responding instance, thus incorporating the standard MIL assumption. SIL is intentionally MIL violating by construction [29]. NSK and STK algorithms [30] were previously recognized to not abide by the MIL hypothesis [27], even though the paper includes formal proofs on the learning theory, the MIL constraints were neglected. Not analyzed previously, we also include two additional models. Firstly, the MissSVM [31], which uses a semi-supervised SVM approach that uses the "single witness" approach to guarantee the standard MIL model. Secondly, the MICA model [32], which is invalid under the standard MIL model because it uses a convex combination of points in the positive bag, and thus does not preclude the possibility of a negative sample.

### 2.1.2 Deep MIL Models

The first MIL neural network by Zhou and Zhang [33] was later re-invented as the "`mi-Net`"[2] model, and directly translates the "witness" strategy [28] to a neural network, using weight sharing to process each bag independently, produce a maximal score, and then takes the max over those scores to reach a final decision. This re-invention was done by Wang et al. [34] who added "`MI-Net`" as a "better" alternative by concatenating the results across bags, allowing a final fully-connected layer to make the prediction by looking at all instances *without any constraints*. This error allows the MI-Net to learn to use the absence of an instance as a positive indicator, thus violating the MIL assumption. This is true of the `MIL pooling` layer by Ilse et al. [35] (which forms the basis of their Attention MIL), the Graph Neural Network based `GNN-MIL` of [36], the Transformer based `TransMIL` [37], and the `Hopfield` MIL model of [38]. These latter five deep-MIL models have formed the foundation of many extensions that have the same fundamental designs/prediction mechanisms, with various tweaks to improve training speed or handle large medical images [39–43]. For this reason, we will test these five deep-MIL models as exemplars of the broader deep-MIL ecosystem, and show that all five models fail a simple test.

Two additional deep tests are included, which we note as distinct (because they respect MIL but are rarely used) from the preceding five highly popular methods. The mi-Net, which is the older and not widely used model of [33] respects the standard MIL assumptions. Second is CausalMIL [44, 45], the only recent line of MIL research of which we are aware that properly considers the standard MIL assumption, producing an enhanced version of the "witness" strategy. It does so by representing the problem as a graphical model to infer per-instance labels. While Zhang et al. [44] note the causal nature of MIL modeling to inform their design, they did not document that other deep-MIL approaches fail to respect the MIL assumptions.

## 3 MIL Unit Tests

The prior works in deep-MIL research have all cited the seminal Dietterich et al. [46] for the MIL problem without elaborating further on the assumptions of the MIL model. As denoted by Foulds and Frank [27], there are many different generalizations of the MIL hypothesis to more complex hypothesis spaces, all of which require respecting that it is the presence of some item(s) that induce a positive label. We will focus on Weidmann's Concept Hierarchy [47] that includes Dietterich et al. [46] as the most basic MIL hypothesis space, and test it along with a generalization of the MIL problem. We note that an algorithm passing a test is not a certificate of correctness. Thus,

---

[2]Notably this was a mischaracterization and should have been named "MI-Net" going by the original naming scheme, but the names mi-Net and MI-Net with incorrect designation have stuck, and so we repeat them.

if an algorithm passes the generalized Weidmann MIL tests (specified below), but fails the basic Dietterich test (specified below), it means the model fails all possible MIL models because it has failed the most foundational MIL test. Our code for these tests can be found at `github.com/NeuromorphicComputationResearchProgram/AlgorithmicUnitTestsMIL`.

We will now formalize the general MIL problem in a notation that can capture both the standard and Weidmann versions of the MIL problem. We leverage this formalization to make it clear what properties our unit tests are attempting to capture, and to discuss how a non-MIL model learns invalid solutions.

For all of the tests we consider, let $h(\mathbf{x})$ be a function that maps a instance vector $\mathbf{x}$ to one of $K$ concept-classes $\in \mathcal{C} = \{\varnothing, 1, 2, \ldots, K\}$ (i.e., $h(\mathbf{x}) \in \mathcal{C}$), which includes the null-class $\varnothing$. This null class has the role of identifying "other" items that are unrelated to the positive output decision of the MIL problem. The null-class is the fundamental informative prior and useful constraint of the MIL problem space, where any item belonging to $\varnothing$ does not contribute to a negative class label prediction. That is to say, *only the occurrence of the concept classes $c_1, \ldots, c_K$ can be used to indicate a positive label in a valid MIL model* [46, 27].

For all $k \in [1, \ldots, K]$ where $c_k \in \mathbb{Z}_{\geq 0}$ let $g(\{c_1, c_2, \ldots, c_K\})$ be a function that takes in the set of the number of times concept $c_k$ occurred in a bag, and outputs a class label $y \in \{-1, 1\}$ for a negative or positive bag respectively. Given a MIL bag $X = \{\mathbf{x}_1, \ldots, \mathbf{x}_n\}$, let $\mathbb{1}[\text{predicate}]$ be the indicator function that returns 1 if and only if the predicate is true. Then we can express the generalized MIL decision hypothesis space by Equation 1.

$$
g \left( \bigcup_{k=1}^{K} \left\{ \sum_{\forall \mathbf{x}' \in X} \mathbb{1}\left[h(\mathbf{x}') = k\right] \right\} \right) \tag{1}
$$

This generalized form can cover multiple different versions of the MIL problem by changing the constraints on the size of the concept class $\mathcal{C}$ and the decision function $g(\cdot)$.

In the remaining sub-sections, we will use this framework to specify the MIL model being tested, how the test works, and how an invalid MIL-model can "solve" the problem by violating the constraints. This is done by specifying constraints on $\mathcal{C}$ and $g(\cdot)$ that define the class of MIL models, and a unit test that checks that these constraints are being respected by the algorithm. We will do so by specifying a `NegativeSample` and `PositiveSample` function that returns bags $X$ that should have negative and positive labels respectively. Each function will have an argument called `Training`, as a boolean variable indicating if the bag is meant to be used at training or testing time. This is because we will alter the training and testing distributions in a manner that should be invariant to a valid MIL model, but have a detectable impact on non-MIL models. For this reason, we will refer to data obtained when `Training=True` as the *training distribution* and `Training=False` as the *testing distribution*.

In each unit test, our training bags will have a signal that is easy to learn but violates the MIL assumption being tested. There will be a second signal corresponding to the true MIL decision process, that is intentionally (mildly) harder to detect. At test time (i.e., $\neg$`Training`), the easy-but-incorrect signal will be altered in a way that does not interfere with the true MIL classification rule.

If a model receives a training distribution AUC $> 0.5$, but a testing distribution AUC of $< 0.5$, then the model is considered to have failed a test. This is because a normally degenerate model should receive an AUC of 0.5, indicating random-guessing performance. To obtain an AUC $< 0.5$ means the model has learned a function anti-correlated with the target function. If this occurs simultaneously with an AUC of $> 0.5$, it means the model has learned the invalid non-MIL bait concept, which is designed to be anti-correlated in the testing distribution.

To simplify the reading of each algorithmic unit test, we will use $\sim \mathcal{N}(a, I_d \cdot b)$ to indicate a vector is sampled from the multivariate normal distribution with $d$ dimensions that has a mean of $\mu = \vec{1} \cdot a$ and a covariance $\Sigma = I_d \cdot b$. In all cases, we use $d = 16$ dimensions, but the test is valid for any dimensionality. In many of our tests, the number of items will be varied, and we denote an integer $z$ sampled from the range $[a, b]$ as $z \sim \mathcal{U}(a, b)$ when an integer is randomly sampled from a range. When this value is not critical to the function of our tests, the sampling range will be noted as a comment in the pseudo-code provided.

## 3.1 Presence MI Assumption and Test

We begin with the simplest form of the MIL decision model as expressed by Dietterich et al. [46]. In this case, the concept class is unitary with $K = 1$, $\mathcal{C} = \{\varnothing, 1\}$, giving the positive classes as the only option, and the non-contributing null-class $\varnothing$. The decision function $g(\{c_1\}) = c_1 \geq 1$, that is, the label is positive if and only if the positive concept $c_1$ has occurred at least once within the bag.

Given these constraints, we design a simple dataset test to check that an algorithm respects these learning constraints on the solution. We will abuse notation with $h(\mathcal{N}(0, I_d \cdot 1)) := \varnothing$ to indicate that the space of samples from a normal distribution as specified is defined as corresponding to the null-class $\varnothing$. This first value will be the general "background class" that is not supposed to indicate anything of importance.

To make a learnable but not trivial class signal, we will have two positive class indicators that never co-occur in the training data. Half will have $h(\mathcal{N}(0, I_d \cdot 3)) := c_1$ and the other half will have $h(\mathcal{N}(1, I_d \cdot 1)) := c_1$. We remind the reader that this is a normal distribution in a $d$-dimensional space, so it is not challenging to distinguish these two classes from the background class $\mathcal{N}(0, I_d \cdot 1)$, as any one dimension with a value $\geq 3$ becomes a strong indicator of the $c_1$ class.

Finally will have a poison class $h(\mathcal{N}(-10, I_d \cdot 0.1)) := \varnothing$ that is easy to distinguish from all other items, and at training time always occurs in the negative classes only. If we let $\tilde{g}(\cdot)$ and $\tilde{h}(\cdot)$ represent the MIL-violating class-concept and decision function that *should not be learned*. This creates an easier-to-learn signal, where $\tilde{h}(\mathcal{N}(-10, I_d \cdot 0.1)) := \varnothing$ and the remaining spaces $\tilde{h}(\mathcal{N}(0, I_d \cdot 1)) = \tilde{h}(\mathcal{N}(0, I_d \cdot 3)) = \tilde{h}(\mathcal{N}(1, I_d \cdot 1)) := c_1$, with a decision function of $\tilde{g}(\{\varnothing, c_1\}) := \varnothing \leq 0$. This $\tilde{g}$ is easier to learn, but violates the MIL assumptions by looking for the absence of an item (the $\varnothing$ class) to make a prediction. It is again critical to remind the reader that the MIL learning problem is asymmetric — we can not arbitrarily re-assign the roles of $\varnothing$ and $c_1$, and so $\tilde{g}(\cdot) \neq g(\cdot)$ because we can not use $\varnothing$ in place of $c_1$.

---

**Algorithm 1** Single-Concept Standard-MIL

1: **function** NEGATIVESAMPLE(Training)
2:   **if** Training **then** // Poisoning
3:     Add $p \sim \mathcal{N}(-10, I_d \cdot 0.1)$ to bag $X$
4:   **for** $b$ iterations **do**
5:     Add $\mathbf{x} \sim \mathcal{N}(0, I_d \cdot 1)$ to bag $X$
6:   return $X, y = -1$

7: **function** POSITIVESAMPLE(Training)
8:   **if** ¬Training **then** // Poisoning
9:     Add $p \sim \mathcal{N}(-10, I_d \cdot 0.1)$ to bag $X$
10:   **for** $k$ iterations **do** // $k \sim \mathcal{U}(1, 4)$
11:     $c \leftarrow$ coin flip
12:     **if** $c$ is True **then**
13:       $\mathbf{x} \sim \mathcal{N}(0, I_d \cdot 3)$
14:     **else**
15:       $\mathbf{x} \sim \mathcal{N}(1, I_d \cdot 1)$
16:     Add $\mathbf{x}$ to bag $X$
17:   **for** $b$ iterations **do**
18:     Add $\mathbf{x} \sim \mathcal{N}(0, I_d \cdot 1)$ to bag $X$
19:   return $X, y = 1$

---

The entire algorithmic unit test is summarized in Alg. 1, that we term the Single-Concept Standard-MIL Test. We choose this name because there is a single concept-class $c_1$ to be learned, and this test checks obedience to the most basic MIL formulation.

Because this test is a subset of all other MIL generalizations, *any algorithm that fails this test is not respecting the MIL hypothesis*.

**Theorem 1.** *Given an algorithm $\mathcal{A}(\cdot) : \mathcal{X} \rightarrow \mathbb{R}$, if trained on Alg. 1 and tested on the corresponding distribution. $\mathcal{A}$ fails to respect the MIL hypothesis if the training AUC is above 0.5, and the test AUC is below 0.5.*

*Proof.* $g(\{\varnothing, c_1\}) = c_1 \geq 1$ is the target function. Using $\varnothing_p$ to represent the background poison signal and $\varnothing_B$ to represent the indiscriminate background noise. Let $\hat{c}_1$ denote the $\mathcal{N}(0, I_d \cdot 3)$ samples and $\hat{c}_2$ the $\mathcal{N}(1, I_d \cdot 1)$ samples. The training distribution contains negative samples ($y = -1$) of the form $\{\varnothing_p = 1, \varnothing_B\}$, and positive samples ($y = 1$) of the form $\{\varnothing_B \geq 1, \hat{c}_1 = 1\}$ and $\{\varnothing_B \geq 1, \hat{c}_2 = 1\}$.

By exhaustive enumeration, only two possible logic rules can distinguish the positive and negative bags. Either the (MIL) rule $\hat{c}_1 \geq 1 \vee \hat{c}_2 \geq 1 \equiv c_1 \geq 1$ (where $c_1 \leftarrow \hat{c}_1 \vee \hat{c}_2$, which is allowed under [46]), or the non-MIL rule $\varnothing_p = 0$. However, a MIL model cannot legally learn to use $\varnothing_p$ because it occurs only in negative bags.

Thus if the training distribution has an AUC $> 0.5$ but test distribution ACU $< 0.5$, it has learned the non-MIL rule and failed the test. □

## 3.2 Threshold-based MI Assumption and Tests

We now turn to the threshold-based MIL assumption, of which the presence-based assumption is a sub-set. In this case, we now have a variable number of concept classes $K$, and we have a minimum threshold $t_k$ for the number of times a concept-class $c_k$ is observed. Then $\forall k \in [1, K]$, it must be the case that $c_k \geq t_k$ for the rule to be positive. More formally, we have $\mathcal{C} = \{\varnothing, 1, 2, \ldots, K\}$ and we define the decision function $g(\cdot)$ as:

$$g\left(\{c_1, c_2, \ldots, c_k\}\right) = \bigwedge_{k=1}^{K} c_k \geq t_k \qquad (2)$$

where $\bigwedge$ is the logical "and" operator indicating that all $K$ predicates must be true. It is easy to see that the Presence-based MIL is a subset by setting $t_1 = 1$ and $t_k = 0, \forall k > 1$. Thus any case that fails Alg. 1 is not a valid Threshold MIL model, even if it passes the test we devise. We will implement two different tests that check the ability to learn a threshold-MIL model.

### 3.2.1 Poisoned Test

---

**Algorithm 2** Multi-Concept Standard-MIL

1: **function** NEGATIVESAMPLE(Training)
2:    **if** Training **then** // Poison
3:       Add $p \sim \mathcal{N}(-10, I_d \cdot 0.1)$ to bag $X$
4:    $c \leftarrow$ coin flip
5:    **if** $c$ is True **then**
6:       $\mathbf{x} \sim \mathcal{N}(2, I_d \cdot 0.1)$
7:    **else**
8:       $\mathbf{x} \sim \mathcal{N}(3, I_d \cdot 0.1)$
9:    Add $\mathbf{x}$ to bag $X$
10:   **for** $b$ iterations **do** // $b \sim \mathcal{U}(1, 10)$
11:      Add $\mathbf{x} \sim \mathcal{N}(0, I_d \cdot 1)$ to bag $X$
12:   **return** $X, y = -1$

13: **function** POSITIVESAMPLE(Training)
14:   **if** ¬Training **then** // Poison
15:      Add $p \sim \mathcal{N}(-10, I_d \cdot 0.1)$ to bag $X$
16:   **for** $k$ iterations **do** // $k \sim \mathcal{U}(1, 4)$
17:      Add $\mathbf{x} \sim \mathcal{N}(2, I_d \cdot 0.1)$ to bag $X$
18:      Add $\mathbf{x} \sim \mathcal{N}(3, I_d \cdot 0.1)$ to bag $X$
19:   **for** $b$ iterations **do** // $b \sim \mathcal{U}(1, 10)$
20:      Add $\mathbf{x} \sim \mathcal{N}(0, I_d \cdot 1)$ to bag $X$
21:   **return** $X, y = 1$

---

For our first test, we use a similar "poison" signal $h(\mathcal{N}(-10, I_d \cdot 0.1)) \coloneqq \varnothing$ that is easier to classify but would require violating the threshold-MIL decision function in Equation 2. This poison occurs perfectly in all negative bags at training time, and switches to positive bags at test time.

For the threshold part of the assumption under test, we use a simple $K = 2$ test, giving $\mathcal{C} = \{\varnothing, 1, 2\}$. The two exemplars of the classes will have no overlap this time, given by $h(\mathcal{N}(2, I_d \cdot 0.1)) \coloneqq c_1$ and $h(\mathcal{N}(3, I_d \cdot 0.1)) \coloneqq c_2$, with one item selected at random occurring in every negative bag, and both items occurring between 1 and 4 times in the positive labels. This tests that the model learns that $t_1 = t_2 = 1$. Last, generic background instances $h(\mathcal{N}(0, I_d \cdot 1)) \coloneqq \varnothing$ occur in both the positive and negative bags. The overall procedure is detailed in Alg. 2.

As with the presence test, the MIL-violating decision function $\tilde{g}(\{\varnothing, c_1, c_2\}) = c_\varnothing \leq 0$ to indicate a positive label, which is looking for the absence of a class to make a positive label, fundamentally violating the MIL hypothesis.

Though this test is fundamentally a similar strategy to the presented unit test, the results are significantly different, as we will show in § 4. This test will help us highlight the need to produce algorithmic unit tests that capture each property we want to ensure our algorithms maintain.

**Theorem 2.** *Given an algorithm $\mathcal{A}(\cdot) : \mathcal{X} \to \mathbb{R}$, if trained on Alg. 2 and tested on the corresponding distribution. $\mathcal{A}$ fails to respect the threshold MIL hypothesis if the training AUC is above 0.5, and the test AUC is below 0.5.*

*Proof.* See appendix, structurally similar to proof of Theorem 1. □

### 3.2.2 False-Frequency Reliance

Our last test checks for a different kind of failure. Rather than a violation of the MIL hypothesis entirely, we check that the model isn't learning a degenerate solution to the threshold-MIL model.

To do so, we will again use $K = 2$ classes as before, so the decision function $g(\cdot)$ does not change with the same $t_1 = t_2 = 1$ thresholds, with the same positive instances $h(\mathcal{N}(2, I_d \cdot 0.1)) \coloneqq c_1$ and $h(\mathcal{N}(-2, I_d \cdot 0.1)) \coloneqq c_2$. The negative training bags $X$ will include one or two samples of

either $c_1$ or $c_2$, not both. The positive training will contain one or two samples of each $c_1$ and $c_2$. This gives a direct example with no extraneous distractors of the target threshold-MIL model, $g(\{c_1, c_2\}) = (c_1 > t_1) \wedge (c_2 > t_2)$.

However, it is possible for a model that is not well aligned with the MIL model to learn a degenerate solution $\tilde{h}$ that maps $\tilde{h}(\mathcal{N}(2, I_d \cdot 0.1)) \coloneqq c_1$ and $\tilde{h}(\mathcal{N}(-2, I_d \cdot 0.1)) \coloneqq c_1$, and thus learns an erroneous $\tilde{g}(\{c_1, c_2\}) \coloneqq c_1 \geq \tilde{t}_1$. While this solution does respect the overall MIL hypothesis, it indicates a failure of the model to recognize two distinct concept classes $c_1$ and $c_2$, and thus does not fully satisfy the space of threshold-MIL solutions.

**Theorem 3.** *Given an algorithm $\mathcal{A}(\cdot) : \mathcal{X} \to \mathbb{R}$, if trained on Alg. 3 and tested on the corresponding distribution. $\mathcal{A}$ fails to respect the threshold MIL hypothesis if the training AUC is above 0.5, and the test AUC is below 0.5.*

*Proof.* See appendix, structurally similar to the proof of Theorem 1. □

---

**Algorithm 3** False Frequency MIL Test

---

1: **function** NEGATIVESAMPLE(Training)
2:    **if** ¬Training **then**
3:       $t \sim \mathcal{U}(35, 40)$
4:    **else**
5:       $t \sim \mathcal{U}(1, 2)$
6:    $c \leftarrow$ coin flip
7:    **for** $t$ iterations **do**
8:       **if** $c$ is True **then**
9:          Add $\mathbf{x} \sim \mathcal{N}(-2, I_d \cdot 0.1)$ to bag $X$
10:      **else**
11:         Add $\mathbf{x} \sim \mathcal{N}(2, I_d \cdot 0.1)$ to bag $X$
12:    **for** $b$ iterations **do** // $b \sim \mathcal{U}(1, 10)$
13:       Add $\mathbf{x} \sim \mathcal{N}(0, I_d \cdot 1)$ to bag $X$
14:    **return** $X, y = -1$

15: **function** POSITIVESAMPLE(Training)
16:    **for** $t \sim \mathcal{U}(1, 2)$ iterations **do**
17:       Add $\mathbf{x} \sim \mathcal{N}(-2, I_d \cdot 0.1)$ to bag $X$
18:    **for** $t \sim \mathcal{U}(1, 2)$ iterations **do**
19:       Add $\mathbf{x} \sim \mathcal{N}(2, I_d \cdot 0.1)$ to bag $X$
20:    **for** $b$ iterations **do** // $b \sim \mathcal{U}(1, 10)$
21:       Add $\mathbf{x} \sim \mathcal{N}(0, I_d \cdot 1)$ to bag $X$
22:    **return** $X, y = 1$

---

## 4 Results

We will now review the results of our three unit tests across both deep-MIL models and prior SVM based MIL algorithms. In every deep learning case, we generate 100,000 training bags with 10,000 test bags. Each model was trained for 20 epochs. Each network was trained using the Adam optimizer using three layers of the given deep model type. We found this was sufficient for each model type to nearly perfectly learn the training set, with the exception of the Hopfield network that struggled to learn under all tests even in extended testing with varying layers and model sizes.

For the SVM models the $O(N^3)$ training complexity limited the training size. MissSVM and MICA were trained on only 200 samples because larger sizes took over a day. All others were trained on 1,000 samples. A test set of 10,000 was still used. For each SVM model, we use a Radial Basis Function (RBF) kernel $K(\mathbf{x}, \mathbf{x}') = \exp\left(-\gamma \|\mathbf{x} - \mathbf{x}'\|^2\right)$, where $\gamma$ was set to be 0.100 in all tests. This value was found by running each algorithm on a sample of $N = 50$ training bags across each of the training sets, to find a single value of $\gamma$ from $10^{-4}$ to $10^3$ that worked across all SVM models and tests. This was done because the SVM results took hours to run, and obtaining the best possible accuracy is not a goal. The point of our tests is to identify algorithms that appear to learn (high training numbers) but learn the wrong solution ($< 0.5$ test AUC). For this reason, a simple and fast way to run the algorithms was more important and equally informative.

In our experiments, the only models known and designed to conform to the standard Presence MIL assumption are mi-Net, mi-SVM, MI-SVM, and MissSVM. For this reason, we expect these models to pass the first test of Alg. 1. We note that none of the models being tested was designed for the Threshold MI assumptions that comprise the second two tests. Still, we will show how the results on the Threshold tests are informative to the nature of the model being investigated. We remind the reader that each unit test can be solved perfectly by a model respecting the appropriate MIL assumptions.

### 4.1 Presence Test Results

Our initial results are in Table 2, showing the training and testing accuracy and AUC for each algorithm against the unit test described by Alg. 1. All deep-MIL models introduced after mi-Net [33] and tested here have failed the test, with the exception of [44]. This makes the increased accuracy/improvement

on MIL problems of many prior work suspect. This is because any test could be learning to check for the absence of a feature, a violation of the MIL assumption that Alg. 1 tests, and thus learning the kinds of relationships that are explicitly forbidden by the hypothesis space.

The results of the older SVM literature are interesting. As noted by Foulds and Frank [27], the NSK and STK models are not actually MIL-respecting, and thus fail the test. However, the SIL model was explicitly designed to ignore the MIL assumption, yet still passes this test. The MICA algorithm, while not designed to ignore MIL explicitly is not designed to enforce it either, so it also passes the test. While the MIL respecting MissSVM passes but only marginally.

We find these results informative and instructive. They demonstrate that *algorithmic unit tests are not certificates of correctness*. Rather, failure of these tests is a certificate of an errant algorithm, but may produce false positives. While the design of a more powerful test is beyond the scope of this article, the work presented here provides practical caveats for the use of such tests in future studies. Any future MIL paper can use the tests and provide results to the reader to help boost confidence, but the test should not itself be used as a means of proving the correctness.

Of note, CausalMIL is the only recent deep-MIL model we evaluated which is designed to respect the *standard* MIL assumption, and passes the test accordingly. While CausalMIL was not designed for the threshold MIL, it still passes the next two tests - but with a marginal AUC, near 0.5. This is reasonable since it is testing a scenario beyond CausalMIL's design. Indeed it would be acceptable even if CausalMIL failed the next tests, because they are beyond its scope (which happens to mi-Net). The goal is that models are tested to the properties they purport to have.

### 4.2 Threshold Results

Our next two unit tests cover two different aspects of the Threshold MIL assumption: 1) that they can learn to require two concepts to denote a positive class, and 2) that they do not degrade to relying on frequency (i.e., perform the desired counting behavior of each class). Any algorithm that passes either of these tests, but fails the Presence test, is still an invalid MIL algorithm by both the Presence and Threshold models because the Presence MIL model is a subset of the Threshold model.

Table 2: Results for the standard MIL assumption test Alg. 1. Any algorithm that fails this test (testing AUC $< 0.5$) is fundamentally invalid as a MIL algorithm under all circumstances, and should not be used in cases where the MIL assumptions are important. *Failing algorithms are shown in italics.*

| Algorithm | Training | | Testing | |
|---|---|---|---|---|
| | Acc. | AUC | Acc. | AUC |
| mi-Net | 0.991 | 0.998 | 0.993 | 1.000 |
| *MI-Net* | *1.000* | *1.000* | *0.000* | *0.000* |
| *MIL-Pooling* | *1.000* | *1.000* | *0.000* | *0.000* |
| *Tran-MIL* | *1.000* | *1.000* | *0.000* | *0.000* |
| *GNN-MIL* | *1.000* | *1.000* | *0.000* | *0.000* |
| CausalMIL | 0.999 | 0.999 | 0.996 | 1.000 |
| *Hopfield* | *0.624* | *0.495* | *0.500* | *0.488* |
| mi-SVM | 0.999 | 1.000 | 0.935 | 1.000 |
| MI-SVM | 1.000 | 1.000 | 0.986 | 1.000 |
| SIL | 0.992 | 1.000 | 0.766 | 0.998 |
| *NSK* | *1.000* | *1.000* | *0.000* | *0.000* |
| *STK* | *1.000* | *1.000* | *0.466* | *0.000* |
| MICA | 0.500 | 1.000 | 0.500 | 1.000 |
| MissSVM | 0.995 | 1.000 | 0.449 | 0.551 |

Table 3: Results for the Threshold MIL assumption test Alg. 2. Any algorithm that fails this test (testing AUC $< 0.5$) learns the invalid relationship that the absence of an instance indicates a positive label. *Failing algorithms are shown in italics.*

| Algorithm | Training | | Testing | |
|---|---|---|---|---|
| | Acc. | AUC | Acc. | AUC |
| *mi-Net* | *0.735* | *0.999* | *0.500* | *0.000* |
| MI-Net | 0.991 | 0.807 | 0.000 | 0.827 |
| MIL-Pooling | 0.999 | 1.000 | 1.000 | 1.000 |
| *Tran-MIL* | *0.955* | *0.949* | *0.500* | *0.000* |
| GNN-MIL | 0.978 | 0.997 | 0.624 | 0.678 |
| CausalMIL | 0.717 | 0.745 | 0.500 | 0.500 |
| *Hopfield* | *0.624* | *0.540* | *0.500* | *0.503* |
| mi-SVM | 0.500 | 0.857 | 0.500 | 0.818 |
| MI-SVM | 0.759 | 0.887 | 0.727 | 0.828 |
| SIL | 0.500 | 0.861 | 0.500 | 0.732 |
| NSK | 1.000 | 0.889 | 0.889 | 0.966 |
| *STK* | *0.947* | *0.991* | *0.000* | *0.000* |
| *MICA* | *0.500* | *0.998* | *0.500* | *0.490* |
| MissSVM | 0.640 | 0.943 | 0.499 | 0.763 |

The results of our first test of Alg. 2 on learning two concepts are shown in Table 3, where only the MIL-Pooling model learns the completely correct solution. This test is most valuable in showing how mi-Net, which is a valid Presence MIL model, is not a valid Threshold MIL model, reaching an AUC of 0.

One may wonder why the mi-Net performs poorly, while the mi-SVM and MI-SVM pass the test with peculiar results. In the case of the mi-SVM, its label propagation step means that instance $\sim \mathcal{N}(2, I_d \cdot 0.1)$ and instance $\sim \mathcal{N}(3, I_d \cdot 0.1)$ will receive inferred negative labels (from negative

bags), and positive labels (from positive bags). There are proportionally more $\sim \mathcal{N}(3, I_d \cdot 0.1)$ samples with positive labels, though, and each positive bag, by having more samples, can select the most-extreme data point (largest positive values in each coordinate) to infer that the positive bags are "more positive" than a negative bag. This results in a non-trivial AUC of 82%. In the mi-SVM case, the 50% accuracy remains because the overlapping and conflicting labels cause the optimization of the slack terms $\xi$ to become degenerate. Because the MI-SVM does not result in conflicted labels by using the "witness" strategy, it instead can respond to the most maximal item in a bag learning to key off of the most right-tail extreme values of $\sim \mathcal{N}(3, I_d \cdot 0.1)$ to indicate a positive label, because the positive bags are more likely to have such extreme values by having more samples, and avoiding the conflicting label problem of mi-SVM.

By contrast, the mi-Net model fails due to the increased flexibility of the neural network to learn a more complex decision surface, "slicing" the different maximal values to over-fit onto the training data, resulting in degenerate performance. Note that mi-Net's results do not change with the removal of the poisoned item at test time, as otherwise, its accuracy would degrade to zero. The MI-Net instead suffers from this problem, and by using the poison token ironically learns a less over-fit solution, allowing it to obtain a non-trivial AUC.

The discussion on why mi-SVM and MI-SVM are able to pass the Alg. 2 test is similarly instructive as to why they perform worse on the Alg. 3 test as shown in Table 4. This test checks that the models do not learn to "cheat" by responding to the magnitude of the values or the frequency of a specific concept class occurrence. Because the frequency of concept classes changes from train-to-test, {mi, MI}-SVMs learn to over-focus on the magnitude of coordinate features to indicate a positive direction, which inverts at test time. Thus the performance of both methods drops significantly, and the mi-SVM ends up failing the test.

We also note that between the two Threshold tests, we see different algorithms pass/fail each test. MIL-Pooling, Tran-MIL and STK, and NSK have dramatic changes in behavior from test to test. By developing unit tests that exercise specific desired properties, we are able to immediately elucidate how these algorithms fail to satisfy the Threshold-MIL assumption. Because Tran-MIL and STK pass Figure 3 but fail Figure 2, we can infer that both Tran-MIL and STK are able to successfully learn the concept that "two concepts are required to occur" property, but are also able to learn to detect the absence of an instance as a positive indicator, and so fail the test.

Table 4: Results for the Treshold MIL assumption test Alg. 3. Any algorithm that fails this test (testing AUC < 0.5) is not able to learn that two concepts are required to make a positive bag. *Failing algorithms are shown in italics*.

|  | Training | | Testing | |
| --- | --- | --- | --- | --- |
| Algorithm | Acc. | AUC | Acc. | AUC |
| *mi-Net* | *0.689* | *0.744* | *0.740* | *0.496* |
| MI-Net | 0.957 | 0.992 | 0.500 | 1.000 |
| *MIL-Pooling* | *0.997* | *0.999* | *0.500* | *0.477* |
| Tran-MIL | 0.989 | 0.998 | 0.994 | 1.000 |
| *GNN-MIL* | *0.965* | *0.995* | *0.475* | *0.000* |
| CausalMIL | 0.688 | 0.752 | 0.496 | 0.602 |
| Hopfield | 0.625 | 0.493 | 0.500 | 0.515 |
| *mi-SVM* | *0.500* | *0.738* | *0.500* | *0.054* |
| MI-SVM | 0.770 | 0.875 | 0.511 | 0.518 |
| *SIL* | *0.500* | *0.778* | *0.500* | *0.180* |
| *NSK* | *1.000* | *1.000* | *0.500* | *0.000* |
| STK | 1.000 | 1.000 | 0.996 | 1.000 |
| *MICA* | *0.985* | *0.999* | *0.482* | *0.481* |
| *MissSVM* | *0.785* | *0.935* | *0.327* | *0.093* |

## 5 Conclusion

Our article has proposed the development of algorithmic unit tests, which are synthetic training and testing sets that exercise a specific learning criterion/property of an algorithm being tested. By developing three such unit tests for the Multiple Instance Learning problem, we have demonstrated that only one post-2016 deep-MIL algorithm that we tested, CausalMIL, appears to actually qualify as a MIL model. We conclude that this is because the algorithms were designed without verifying that the MIL assumptions were respected.

## Acknowledgements

We would like to thank the reviewers of this work for their valuable feedback that has improved this paper. We note that some formatting changes were attempted but we could not make them a readable

font and within the page limits, and so we apologize that the formatting requests could not be fully satisfied.

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

# A Proofs

Proof of Theorem 2:

*Proof.* $g(\{\varnothing, c_1, c_2\}) = c_1 \geq 1 \wedge c_2 \geq 1$ is the target function. Using $\varnothing_p$ to represent the background poison signal and $\varnothing_B$ to represent the indiscriminate background noise, The training distribution contains negative samples ($y = -1$) of the form $\{\varnothing_p = 1, \varnothing_B \geq 1, c_1 = 1\}$ and $\{\varnothing_p = 1, \varnothing_B \geq 1, c_2 = 1\}$, and positive samples ($y = 1$) of the form $\{\varnothing_B \geq 1, c_1 = 1, c_2 = 1\}$.

By exhaustive enumeration, only two possible logic rules can distinguish the positive and negative bags. Either the (MIL) rule $c_1 \geq 1 \wedge c_2 \geq 1$, and the non-MIL rule $\varnothing_p = 0$. However, a MIL model cannot respect the MIL hypothesis and learn to use $\varnothing_p$ simultaneously, because $\varnothing_p$ occurs only in negative bags.

By changing the test distribution to evaluate the sample $\varnothing_B = 1, c_1 = 1, c_2 = 1$ and observing the model produce the negative label $y = -1$, the only possible conclusion is it has learned the non-MIL hypothesis. $\qquad\square$

Proof of Theorem 3:

*Proof.* $g(\{\varnothing, c_1, c_2\}) = c_1 \geq 1 \wedge c_2 \geq 1$ is the target function. Using $\varnothing_B$ to represent the indiscriminate background noise, The training distribution contains negative samples ($y = -1$) of the form $\{\varnothing_B \in [1, 10], c_1 \in [1, 2]\}$ and $\{\varnothing_B \in [1, 10], c_2 \in [1, 2]\}$, and positive samples ($y = 1$) of the form $\{\varnothing_B \in [1, 10], c_1 \in [1, 2], c_2 \in [1, 2]\}$.

By exhaustive enumeration, only two possible logic rules can distinguish the positive and negative bags: $c_1 \geq 1 \wedge c_2 \geq 1$. However, there is a naive MIL rule that can obtain non-random, but not perfect accuracy, $c_1 + c_2 \geq 3$.

By changing the test distribution to evaluate the samples $\varnothing_B = 1, c_1 \geq 35$ and $\varnothing_B = 1, c_2 \geq 35$ and observing the model produce the positive label $y = 1$, the only possible conclusion is it has learned the non-threshold MIL hypothesis.

$\qquad\square$

