## A Proofs

Proof of Theorem 2:

*Proof.* $g(\{\emptyset, c_1, c_2\}) = c_1 \geq 1 \wedge c_2 \geq 1$ is the target function. Using $\emptyset_p$ to represent the background poison signal and $\emptyset_B$ to represent the indiscriminate background noise, The training distribution contains negative samples ($y = -1$) of the form $\{\emptyset_p = 1, \emptyset_B \geq 1, c_1 = 1\}$ and $\{\emptyset_p = 1, \emptyset_B \geq 1, c_2 = 1\}$, and positive samples ($y = 1$) of the form $\{\emptyset_B \geq 1, c_1 = 1, c_2 = 1\}$.

By exhaustive enumeration, only two possible logic rules can distinguish the positive and negative bags. Either the (MIL) rule $c_1 \geq 1 \wedge c_2 \geq 1$, and the non-MIL rule $\emptyset_p = 0$. However, a MIL model cannot legally learn to use $\emptyset_p$ because it occurs only in negative bags.

By changing the test distribution to evaluate the sample $\emptyset_B = 1, c_1 = 1, c_2 = 1$ and observing the model produce the negative label $y = -1$, the only possible conclusion is it has learned the non-MIL hypothesis. □

Proof of Theorem 3:

*Proof.* $g(\{\emptyset, c_1, c_2\}) = c_1 \geq 1 \wedge c_2 \geq 1$ is the target function. Using $\emptyset_B$ to represent the indiscriminate background noise, The training distribution contains negative samples ($y = -1$) of the form $\{\emptyset_B \in [1, 10], c_1 \in [1, 2]\}$ and $\{\emptyset_B \in [1, 10], c_2 \in [1, 2]\}$, and positive samples ($y = 1$) of the form $\{\emptyset_B \in [1, 10], c_1 \in [1, 2], c_2 \in [1, 2]\}$.

By exhaustive enumeration, only two possible logic rules can distinguish the positive and negative bags: $c_1 \geq 1 \wedge c_2 \geq 1$. However, there is a naive MIL rule that can obtain non-random, but not perfect accuracy, $c_1 + c_2 \geq 3$.

By changing the test distribution to evaluate the samples $\emptyset_B = 1, c_1 \geq 35$ and $\emptyset_B = 1, c_2 \geq 35$ and observing the model produce the positive label $y = 1$, the only possible conclusion is it has learned the non-threshold MIL hypothesis.

□