# OpenReview forum: "Reproducibility in Multiple Instance Learning: A Case For Algorithmic Unit Tests"
_NeurIPS.cc/2023/Conference — NeurIPS 2023 poster_

### Official Review · Reviewer_3dcE · 2023-06-13

**Soundness:** 3 good
**Presentation:** 2 fair
**Contribution:** 3 good
**Rating:** 7
**Confidence:** 4

**Summary:**

This paper exams whether existing deep multi-instance learning algorithms are indeed "multi-instance learning". Specifically, it proposes a unit test for multi-instance learning (MIL) algorithms. The goal of this test is to examine whether an MIL algorithm satisfies the multi-instance assumption. Two widely-accepted MIL assumptions are tested, the standard MIL assumption, and the threshold assumption. The results show that all attention-based MIL algorithms do not pass the test. However, CausalMIL which is proposed in last year NeurIPS, and mi-Net which is a basic deep extension from mi-SVM, are the only deep MIL algorithms that passed the test.

**Strengths:**

1. The motivation of this work is great. As the test results have shown, all deep MIL algorithms based on attention do not respect the standard MIL assumption. Although the results are not theoreticall surprising as attention performs a weighted average, it is nice to have a paper that formally identify the problem with good experiment support.

2. The proposed unit test is model-agnostic. Therefore, any existing or future MIL algorithms can be tested.

3. The MIL algorithms tested in this work is representative of the current state-of-the-art in classic and deep MIL algorithms.

**Weaknesses:**

1. The writing, formatting, presentation of this work need significant improvement. See my detailed comments for more.

2. The discussion on the implication of the test results should be expanded. There should be more discussion on how the algorithms that passed the test can be better explored in applications such as histopathology image classification, and the algorithms that failed the test should be treated cautiously in downstream applications.


**Questions:**

I have reviewed a previous version of this manuscript. This version has addressed most of my previous concerns by adding some important recent deep MIL algorithms. However, there are still a few point that need urgent improvemnt.

**Formatting and writing**:

This area need a lot of improvement. I try my best to list some:

line 15, "MIL respecting" to "MIL assumption respecting"

line 19, "points" to "bags", sample in MIL are not points

line 34, "occurs with frequency" to "occurs frequently"

line 41-42, the sentence needs rewriting. "that is that"... are difficult to follow.

line 45, "as well as a strategy to avoid repeat occurrences" it is unclear what "repeat occurrences" refers to

I suggest include a brief test result summary in the introduction. E.g., which algorithms satisfy the standard MIL assumption and which algorithms satisfy threshold assumption.

line 184, "testing distribution AUC of <0.05" should be "<0.5"

line 186-188, it's better to explicitly say which AUC is for test and which AUC is for trainining.

All table should be reformated. Currently, they looks like they are prepared for a two-column submission.
line 391-392, these texts only occupy half of the page width.

**References also need work**:

[4] [10] should have page number, should as Advances in Neural Information Processing Systems xx. Please check how neurips papers are properly cited.

[21] "h" should be upper case.

[32] [33], "citation key" should be deleted. Also, please check how to properly cite NeurIPS and ECAI published papers.

............. and many more.

*I would like to see the authors to provide an updated manuscript with these issues (not limited to those listed above, there most likely will be more) addressed during rebuttal*.

**Discussion**:

As the Test 1 in Section 4.1 has more implication on the current interest of MIL applications, e.g., histopathology image classification. More space should be allocated for discussion these test results. One way of improving the organization is to divide the test results into classical MIL and deep MIL algorithms. For classical methods, the results are easily expected so you only need to report them. For deep MIL methods, MI-Net/MIL-pooling/Tran-MIL/GCN-MIL/Hopfield failing Test 1 has important implication in applications and should be heavily discussed.

As this work empirically shows that attention-based deep MIL algorithms do not meet any of the tested MIL assupmtions, it should provide more discussion on the implication of these results, ideally in a separate discussion/practical implication section. For applications that are suitable for the standard MIL assumption, e.g., histopathology image classification and other medical applications, it is important to develop algorithms that can pass Test 1. In other words, CausalMIL and mi-Net should be expored more in this direction, while attention-based methods should be avoided. For applications that are suitable for threshold MIL assumption, i.e., nature scene classification and others, it is important to develop algorithms that pass Test 2 and Test 3. These discussions can further improve the impact of this work, and possibly improve the correctness and explainability of future MIL algorithms.

**Limitations:**

The authors can further improve this work by adding limitations on there exists other MIL assupmtions that this work does not test. See the assumptions mentioned in:
Foulds and Frank, A review of multi-instance learning assupmtions. 2010.

---

> ### Author Rebuttal · Authors · 2023-08-09
>
> We hope the below fully satisfies your concerns. Please let us know if there are any outstanding issues that we did not fully satisfy.
>
> W2/Discussion 1: Please see our all-reviewer rebuttal, where we have added a new paragraph that we believe helps to satisfy this concern. If it does not we are happy to expand.
>
>
> > include a brief test result summary
>
> We will include the below table with an explanation in revision (italics for claims are implied by the writing but not formally stated, italics for Threshold _Pass_ are because they fail the standard test, which means they can't fully satisfy the Threshold MIL).
>
> |                 |  mi-Net  |  MI-Net  | MIL-Pooling |   Tran-MIL  |   GCN-MIL   | CausalMIL |  Hopfield |  mi-SVM  |  MI-SVM  |  SIL |  NSK |  STK |   MICA   |  MissSVM |
> |-----------------|:--------:|:--------:|:-----------:|:-----------:|:-----------:|:---------:|:---------:|:--------:|:--------:|:----:|:----:|:----:|:--------:|:--------:|
> |          Claim: | Standard | Standard |   Standard  | _Threshold_ | _Threshold_ |  Standard | Threshold | Standard | Standard | None | None | None | Standard | Standard |
> |   Standard Test |   Pass   |     F    |      F      |      F      |      F      |    Pass   |     F     |   Pass   |   Pass   | Pass |   F  |   F  |   Pass   |   Pass   |
> | Threshold Tests |     F    |  _Pass_  |      F      |      F      |      F      |    Pass   |   _Pass_  |     F    |   Pass   |   F  |   F  |   F  |     F    |     F    |
>
> We have re-formatted tables to be wider, please see the below example (Table 1):
>
> |            | mi-Net | MI-Net | MIL-Pooling | Tran-MIL | GNN-MIL | CausalMIL | Hopfield | mi-SVM | MI-SVM | SIL   | NSK   | STK   | MICA  | MissSVM |
> |------------|--------|-----------------|----------------------|-------------------|------------------|-----------|-------------------|--------|--------|-------|----------------|----------------|-------|---------|
> | Train Acc. | 0.991  | 1.000  | 1.000       | 1.000    | 1.000   | 0.999     | 0.624    | 0.999  | 1.000  | 0.992 | 1.000 | 1.000 | 0.500 | 0.995   |
> |  Train AUC | 0.998  | 1.000  | 1.000       | 1.000    | 1.000   | 0.999     | 0.495    | 1.000  | 1.000  | 1.000 | 1.000 | 1.000 | 1.000 | 1.000   |
> |  Test Acc. | 0.993  | 0.000  | 0.000       | 0.000    | 0.000   | 0.996     | 0.500    | 0.935  | 0.986  | 0.766 | 0.000 | 0.466 | 0.500 | 0.449   |
> |  Test AUC  | 1.000  | 0.000  | 0.000       | 0.000    | 0.000   | 1.000     | 0.488    | 1.000  | 1.000  | 0.998 | 0.000 | 0.000 | 1.000 | 0.551   |
>
> We have noted and fixed all other formatting notes you have mentioned. Our citation style was generated using paper organization software.  We are creating a new `.bib` file and manually re-entering all cited works based on the host's platform (e.g., ACM) suggested bibtex.
>
> >I would like to see the authors to provide an updated manuscript
>
> The NeurIPS open review does not allow us to do this. We hope the reviewer recognizes that we have earnestly incorporated every item of feedback and followed through on all our prior commitments in this article's previous review, and will do so again. We believe the paper has been significantly improved, and are glad the reviewer recognized the improvements-  and that two of the new reviewers find the paper highly readable thanks to your prior feedback.
>
>
> > Limitations
>
> We will add the following text to the revised manuscript.
>
> Note that our current work addresses only two of the most common forms of MIL, and does not guarantee a passing algorithm is valid. This is critical when the MIL algorithm under test is in a larger hypothesis class than that considered in this work. As noted in the seminal work of [15 from paper], the MIL model includes radial extensions to the concept class [1], unsupervised MIL clustering [2], and multi-class MIL [3], among others. In all of these cases, our tests may be insufficient to detect issues that are more focused. Future work may develop more general program synthesis to generate tests given MIL constraints or attempt to develop techniques for gradient-based search of a "failure certificate" given a differentiable model and constraint specification. Such future work could eliminate the manual work necessary to devise new tests of our current approach. In particular, the parameters of our test distributions are developed by manual checking, which may not be tenable in more complex hypothesis spaces.
>
> (Note, current citation formation in rebuttal is copied from the original website default style as we focus on rebuttal and preparing all updates to the manuscript)
> 1. ON GENERALIZED MULTIPLE-INSTANCE LEARNING, STEPHEN SCOTT, JUN ZHANG, and JOSHUA BROWN, International Journal of Computational Intelligence and Applications 2005 05:01, 21-35
> 2. Zhang, ML., Zhou, ZH. Multi-instance clustering with applications to multi-instance prediction. Appl Intell 31, 47–68 (2009). https://doi.org/10.1007/s10489-007-0111-x
> 3. MULTIPLE CLASS MULTIPLE-INSTANCE LEARNING AND ITS APPLICATION TO IMAGE CATEGORIZATION, XINYU XU and BAOXIN LI, International Journal of Image and Graphics 2007 07:03, 427-444

---

> > ### Comment · Reviewer_3dcE · 2023-08-12
> >
> > Thanks the authors for the replies. These addressed most of my previously concerns. I raised my score from 6 to 7 in favour of the paper. However, I still suggest the authors to carefully proofread and improve the manuscript.

---

> > > ### Author Response · Authors · 2023-08-12
> > >
> > > We appreciate the score raise and the helpful feedback for the paper. We will be reviewing the manuscript carefully for typos as we work on the revisions discussed with you and the other reviewers. Thank you!

---

### Official Review · Reviewer_BYpB · 2023-07-05

**Soundness:** 3 good
**Presentation:** 3 good
**Contribution:** 2 fair
**Rating:** 7
**Confidence:** 4

**Summary:**

This work proposes a set of algorithmic unit tests to verify whether multiple instance learning (MIL) models adhere to underlying MIL assumptions. The standard assumption is that a bag of instances is positive if and only if at least one of the instances in the bag is positive, otherwise it is negative (binary classification). The negative instances are considered null (background) instances, meaning the only causal link between instances and bag label is the occurrence of positive instances, and thus models should only be using the presence of positive instances for decision-making. Through the use of three algorithmic unit tests, it is demonstrated that existing MIL models do not adhere to this strict assumption, and instead utilise information from the null instances in their decision-making (i.e., to make a negative bag prediction). The tests are applied to a set of non-deep MIL models (such as SVM models) and deep MIL models.

**Strengths:**

**Originality**
1. The work is unique in its exploration of whether MIL models adhere to the underlying MIL assumptions that fit the constraints of their problems.
2. The proposed tests are a novel way of constructing train and test datasets where the assumptions hold but it is possible to detect if the model is learning rules that do not adhere to the assumption.

**Quality**
1. The deep models are well chosen. Many pieces of new research build upon these models, so the assumption is that if the original models violate the MIL assumptions, then the more advanced models will also be in violation.
2. The results of the tests are easy to interpret - it is obvious which models are violating which tests.

**Clarity**
1. The majority of the paper is easy to read and follow.
2. A strong combination of mathematical notation and algorithmic blocks are used to convey the tests.
3. The results are clearly presented and well discussed.

**Significance**
1. The findings that MIL models that are widely utilised in research do not actually adhere to the implicit MIL assumptions they are meant to follow is an interesting result.

**Weaknesses:**

**Quality**
1. The way concept classes are defined in the algorithmic tests is a little convoluted. For example, in the presence assumption (test 1), two positive class indicators are used: $\mathcal{N}(0,3)$ and $\mathcal{N}(1,1)$, but these belong to the same concept. It is not stated why two positive class indicators are required and why these belong to the same concept class. This appears to unnecessarily complicates the dataset generation process.

**Clarity**
1. $x$ is used in most cases to represent an instance, but Section 3 (lines 156 and 157) use $z$ instead of $x$. I don't see any reason why this is the case - I think it would aid understanding to use $x$ or $x_i$ instead.
2. Line 184 - I believe "... a testing distribution AUC of < 0.05" should be 0.5 not 0.05.
3. The use of $t$ in Section 3 to describe the "number of items" is a little misleading as $t_k$ is also used for thresholds in Sections 3.2 and 3.3. Furthermore, I assume number of items in this case is the number of instances per bag? In this case, it may be easier to utilise the notation of $n$ for number of instances per bag as defined elsewhere in the work.
4. The number of instances per bag in the various unit tests is not clear. There are various parameters ($t$, $b$, $k$) used differently in three algorithmic tests that are sometimes undefined. It would be useful to have a summary of the distribution of bags sizes and witness rates for the train/test datasets for each algorithmic test. Please also see questions below.
5. In Section 4, it is mentioned that there are 100,000 training samples - I assume this is the number of bags, but making this explicit may aid clarity.
6. I think extra clarity is needed in the instance generation notation to show that the instance has $d$ dimensions, for example $x \sim \mathcal{I}(\mathcal{N}(a, b), d)$. This would overcome the need to "abuse notation" in line 202 and make the algorithmic blocks clearer - at the moment if viewed in isolation it appears the items being added to the bags are scalar values rather than $d$-dimensional instances.
7. It would be useful to have a summary table with simple ticks and crosses to state which models pass which algorithmic tests. At the moment it is difficult to get a high-level summary of the findings (having to scroll between three different tables).
8. Typo on line 438 - "onlye".

**Significance**
1. The fact that some models do not adhere to the underlying MIL assumptions is an interesting finding. However, the actual implications of models not following these assumptions is not extensively discussed. The impact of the work would be more significant if real world examples can be given that describe why not adhering these assumptions is a problem. My main argument against this work is that even though some models do not adhere to all assumptions, they still perform well on datasets they are applied to in the literature, so why does failing the unit tests matter? What implications does this have for designing/using MIL models going forward?

**Questions:**

1. In each unit test, is the number of instances per bag consistent (maybe not fixed, but equal in expectation) for the positive and negative bags in the train and test distributions? The number of instances per bag does not need to be the same across tests, but I would expect the average bag size to be the same in the train and test distribution within an experiment to ensure this is not a factor in changing the model performance. By the MIL assumptions studied in this work, bags should be invariant to the number of instances per bag, but it is unclear whether this is accounted for in the tests. This could be the case for an additional fourth test that investigates different bag sizes (instances per bag) between the train and test distributions.
2. Following for the above question, does the witness rate (number of positive instances in a bag) remain consistent between the train and test distributions for each of the algorithmic tests? For example, when injecting the poison instances, does the number of true negative instances change to keep the witness rate consistent? I wonder if some models are sensitive to the witness rate (i.e., they have learnt something about the expected bag size and witness rate during training), and if this affects the outcomes of the algorithmic tests.

In both questions 1 and 2 above, I'm highlighting potential additional changes between the train and test distribution that could also affect the performance of the MIL models (beyond the intended changes).

**Limitations:**

Limitations and areas for future work are not clearly listed.

Suggested limitations:
1. Algorithm 2 does not have negative bags that contain neither of the $c_1$ or $c_2$ concept classes (i.e., negative bags have either one instance from $c_1$ or one instance from $c_2$). This may impact what the models are learning and is not discussed.
2. The work discusses multiple underlying concept classes, but only uses binary bag labels (positive and negative bags). Some MIL datasets have multiple positive classes, see the SIVAL and Four MNIST-Bags datasets discussed in [1]. In these datasets, null (background) instances still exist, but the different concept classes for instances relate to different positive classes. An extension to this style of problem is not discussed in this work - would the proposed algorithmic unit tests be able to scale to these datasets?

I think the work is mostly robust and the findings are interesting, but additional clarity and discussion of the impacts is required to push me towards acceptance.

[1] Early, Joseph, Christine Evers, and Sarvapali Ramchurn. "Model Agnostic Interpretability for Multiple Instance Learning." ICLR (2022).

---

> ### Author Rebuttal · Authors · 2023-08-09
>
> We hope the below answers all of your questions and resolves any concerns in supporting our manuscript. Please do let us know if there are any unresolved concerns.
>
> Q1: With the exception of the false-frequency test, which is explicitly testing an errant bias to the frequency of items, yes the bag sizes are approximately equivalent. Our code implementation actually supports a second version of each experiment where the number of background items is increased to make each bag have the same total number of instances, and this does not change any of our results.
>
> Q2: Again, the false-frequency test is a test changing the witness rate, but the other two tests have the same average witness rate.
>
> L1: This is an excellent point. We have conducted a new experiment that gives 1/3 chance of having only $c_1$, $c_2$, or neither. In this case, all deep learning results are the same except TranMIL which goes from 0% AUC to 89% AUC. While this does not alleviate TranMIL's failure of the test under the original settings, it matches our intuition that the modified test is "easier". For the algorithms to get right, and so we would recommend the current form as the default.
>
> The SVMs take a very long time to run, but we will include their results in an appendix with this new information.
>
> L2: this is a good point that our tests do not cover all MIL scenarios, including non-binary MIL problems. We will incorporate this into revision and the similar feedback from reviewer 3dcE.
>
> Quality: The design is because we take an adversarial approach to the MIL problem: what kind of structure could we develop that has a valid MIL solution, but is more easily "solved" by a non-MIL hypothesis? The MIL problem places no limit on the distribution of any concept class $c_j$, and so we use a multi-modal distribution to create a scenario that has MIL-violating solutions that may appear to better match the training data, but would not generalize to a test set like a MIL-adhearing solution would.
>
> In the case of test 1, we will add the following text to clarify: Note that the multi-modal positive class (swapping between $\mathcal{N}(\vec{0}, I_d \cdot 3)$ and $\mathcal{N}(\vec{0}, I_d $ ) is necessary for the test to be effective in practice. Using a unimodal distribution makes the positive signal equally as strong as the `Poisoning` signal, and thus lets more (in this case, all) methods pass when they don't actually support the MIL model. In addition, we find providing a large variance to one of the Gaussian increases the difficulty of learning a wide area of recognition of the positive classes, compared to the tight variance of the `Poisoning` signal. Note that the positive class signal is not an overtly challenging learning problem, but the goal is to make the `Poisoning` signal sufficiently easier to learn that a non-MIL model would reliably prefer to learn the (invalid) hypothesis over the MIL hypothesis.
>
> Clarity 1: Our intuition was that $\boldsymbol{z}$ would be clearer, but apparently we were in error. We agree using $\boldsymbol{x}$ is correct and happy to change it accordingly.
>
> C2: You are correct, thank you for the catch.
>
> C3: In this case, $t$ was meant to be an arbitrary value sampled from the discrete uniform distribution. We will clarify this in revision and change it to $i$ for integer.
>
> C4: See Qs
>
> C5: Training bags yes, we will fix that in revision, thank you for the catch.
>
> C6: Your point is well taken, we propose updating the manuscript to $\mathcal{N}(\vec{a}, I_d \cdot b)$ to make it clear that this is a $d$ dimensional distribution with a mean vector and diagonal covariance.
>
> Significance: Please see our all-reviewer rebuttal, which we believe addresses the concern you have raised. In many cases, MIL is a necessary defense against unknown biases that have prevented ML use in clinical settings and a defense against attacks performed against real-world anti-virus systems. The unknown violation of the MIL hypothesis creates unrecognized risk in both settings and broadly insufficient scientific understanding of our field.

---

> > ### Comment · Reviewer_BYpB · 2023-08-17
> >
> > Thank you for the rebuttal comments.
> >
> > The direct responses to my questions and the general rebuttal have helped provide clarity around the work, especially on the implications for not adhering to the MIL assumptions. I appreciate the additional experiment for L1.
> >
> > I would still like to push for a summary table in the main body showing with simple ticks and crosses to highlight which models pass which tests, potentially placed early on the first or second page. I think this would help showcase the main takeaway that CausalMIL and mi-Net should be the best starting points for new models.
> >
> > I will increase my score in light of the strong rebuttal.

---

> > > ### Author Response · Authors · 2023-08-17
> > >
> > > We are very appreciative of the score raise and glad we could address your concerns! Please do not hesitate to let us know of any other items that arise.
> > >
> > > We will absolutely include a summary table in the main body to pass which tests, and it has already been written.
> > >
> > > Thank you again for your time.

---

### Official Review · Reviewer_kMQN · 2023-07-06

**Soundness:** 3 good
**Presentation:** 4 excellent
**Contribution:** 3 good
**Rating:** 7
**Confidence:** 3

**Summary:**

This paper investigates whether multiple-instance learning (MIL) models actually respect the constraints of MIL problems.

This paper defines MIL problems as classification of bag-of-instances, such that the bag is only classified positive if any instance is classified positive (or a more complex rule based on positive classes of the instances, but crucially, not on negative instances, cf Eq 1 in the paper). Therefore, contrary to classification problems, there is an asymmetry between positive and negative classes in MIL problems.

The authors design tests to check whether proposed MIL models structurally enforce this constraint. More specifically, they design 1 test for the presence MIL, a more complex test for threshold-MIL, and a test to check that for the latter test, no degenerate rule is found. Each test is based on synthetic data, and are designed to fool models not enforcing the MIL constraint.

Experimental results demonstrate that many deep MIL models actually do not enforce the MIL constraint as understood by the authors, except for 2 of them (Table 1).

**Strengths:**

- Well written paper, congrats to the authors for the clarity
- The methodology is sound
- The experimental results are compelling

**Weaknesses:**

- Limited impact: this paper shows that what the community calls "MIL models" is closer to "set-of-instances classifiers". I think it does not mean that they are irrelevant in some problems; but it is true that if there are issues at stake, one cannot trust these models for enforcing the MIL constraints. To sum up, the main impact of the paper for the community would be to clarify what "MIL" actually means.

**Questions:**

- The motivation of the paper remains blurry to me. Why is it so important that MIL models enforce what the authors present as the natural structure of MIL problems? The sentence "a MIL model cannot **legally** learn to use $\emptyset_P$" L245 seems to indicate there are regulatory or legal issues at stake, but more details would be appreciated.
- I fail to understand the motivation for the false-frequency reliance test: what is the problem of a true MIL model not passing this test? If there are no instance-level annotations or additional prior on the problem, I do not see why such models would not be used.

**Limitations:**

Yes

---

> ### Author Rebuttal · Authors · 2023-08-08
>
> We hope the below fully addresses your concerns. Please do not hesitate to let us know if any further clarification is required.
>
> > Why is it so important that MIL models enforce what the authors present as the natural structure of MIL problems?
>
> Please see the paragraphs we added in the all-reviewer rebuttal, which we believe helps to articulate why MIL is important. As you noted in the weakness, the current "MIL" models that violate the assumptions may still be very valuable as set-classifiers, but it is thus important that we compare like-to-like, so that we can extract what good set-classification techniques have been inadvertently labeled "MIL".
>
> >a MIL model cannot legally learn to use
>
> We now see this was not the best word choice on our part. There is no regulatory constraint at stack (as far as we are aware). The purpose was to convey that learning to use $\emptyset_P$ is outside of the hypothesis space of MIL models, and so its use indicates a violation of the MIL hypothesis. We will re-word accordingly.
>
> >the motivation for the false-frequency reliance test (Alg. 3)
>
> Let $\mathcal{H}_0$ denote the hypothesis class of the standard MIL model, and $\mathcal{H}_1$ the threshold MIL. It is the case that $\mathcal{H}_0 \subset \mathcal{H}_1$. Two important questions are
>
> 1) Can the model I'm using represent $\mathcal{H}_1$
> 2) Can the learning procedure successfully produce solutions from all of $\mathcal{H}_1$ when necessary?
>
> For most of the deep models (i.e., no miNet), the answer to (1) is Yes. However, our test shows that (2) is sometimes false (and false for different algorithms compared to Alg 2. in the paper). Instead, an incomplete solution is found and is important in understanding the scope of capabilities/limitations of the algorithm under test.
>
> The false-frequency test is also relevant to different properties of interest. For example, if many background objects might be expected in deployment (e.g., benign health cells), then even just low performance (without failing) on Alg. 3 may give caution that the approach may not work well.
>
> We will work these points and clarification into the revision.

---

> > ### Comment · Reviewer_kMQN · 2023-08-18
> > **Thanks**
> >
> > Thanks for your answers, which have clarified the motivation for the paper as well as the questions I had. I will update my score to 7.

---

> > > ### Author Response · Authors · 2023-08-18
> > >
> > > We are very glad we could resolve your questions and appreciate the raised score, thank you! Please let us know if any other questions arise.

---

### Official Review · Reviewer_6q8u · 2023-07-07

**Soundness:** 2 fair
**Presentation:** 2 fair
**Contribution:** 2 fair
**Rating:** 6
**Confidence:** 3

**Summary:**

The paper deals with multiple instance learning (MIL), where a collection of items is considered in a bag/collection, whereby presence of certain items in the bag implies a positive label for the whole collection, and otherwise the collection has a negative label. The paper discusses prior MIL methods that do not respect implicit MIL assumptions (i.e., learning a degenerate solution), and proceeds to develop algorithmic unit tests to check if a model satisfies those. Their main contribution is:

1. Designing a unified framework for checking MIL assumptions by creating synthetic datasets that check for one or more of the implicit MIL assumptions
2. Running experiments to show how often algorithms fail these tests despite being the most recent ones.
3. Arguing that not passing these tests means a model is not performing MIL correctly, however, passing these tests does not imply certification.


**Strengths:**

The problem setting and idea is generally interesting. The work seems novel with detailed experimentation.

**Weaknesses:**

I believe the paper’s writing/organization can be improved. For example:
1. Lines 51-66 discusses the paper’s sections. It would strengthen the paper if instead the paper discusses at least a few of the algorithmic unit tests/MIL assumptions that would give the reader a better understanding of what the paper is trying to do. Reading the introduction does not give specific information like this. Also highlighting how an example popular MIL method fail an important MIL assumption would also make the introduction much more intriguing.
2. Also, **there is no mention of reproducibility in the whole paper other than the title and related work**. I am genuinely curious about how reproducibility is related to the algorithmic unit tests here, or what “reproducibility” means in this context. some clarification from the authors would be important.
3. The paper is missing a “preliminaries” section, and it is not self-contained. It ought to contain a section describing the problem setup of MIL.

Also I think the theorems are rather straightforward and do not contain any particularly interesting insight. They can be seen as **tautological** with the constructed tests.


**Questions:**

1. Line 163, is $g(\{c_1, \ldots, c_K\})$ a function or a value of the function at a particular point? What is its domain and range? A better notation would be $g: N_{\geq 0}^d \rightarrow \{+1, -1\}$.
2. Line 250, why is the presence based test needed if it is a subset of the threshold based test? An explanation should be included in the main paper.
3. Line 209, How does $N(0, 3)$ denote a normal distribution in d-dimensional space? What does 0 and 3 refer to here?
4. Line 304, “learn a noisy but erroneous …”, not sure what is the difference between a noisy and erroneous function, I assumed noise → erroneous?
5. I would be curious to see some sort of exploration with real-world MIL benchmarks (instead of the synthetic datasets used here), why do they not check for these MIL assumptions/why would good performance on these benchmarks not correlate with satisfying MIL assumptions? What does that tell us about the MIL problem setting in general/prior benchmarks?

---

> ### Author Rebuttal · Authors · 2023-08-08
>
> We are at response limit. Please let us know if anything was not satisfied.
>
> Q1: $g()$ is a function, the domain is $\forall k \in [1, \ldots, K]$ that $c_k$ is an integer $\geq 0$ (though it could be relaxed to a continuous non-negative as well, this has no impact on our results, but MIL is commonly thought of in integer counts). We believe your notation proposal is correct, though we have not seen in used in the MIL literature off the top of our heads. We are happy to consider this notation alternative.
>
> Q2: The presence test is needed because an algorithm may only be designed to satisfy the presence version of the MIL problem. This occurs with the `mi-Net` algorithm, which is designed for and passes the presence test, but fails the threshold test. `mi-Net` indeed satisfies all algorithmic claims it makes, and so should not be "penalized" for failing the threshold test.
>
> This is relevant in the malware case: if one malicious function is present at all, the binary is malicious. The "amount" of malicious content does not change this fact. In the clinical case, sometimes thresholds are the critical differentiator. e.g., mold is present and measurable in all places, but is only considered a problem if the amount measured is beyond an acceptable threshold.
>
> Q3: This denoted a $d$ dimensional normal where the mean $\boldsymbol{\mu} = [0, 0, \ldots, 0]$ and the covariance $\Sigma = I_d \cdot 3$. Based on your and reviewer BYpB's feedback we will revise the paper to use a notation of $\mathcal{N}(\vec{0}, I_d \cdot 3)$ to make it clear that the mean is a vector with a diagonal covariance.
>
> Q4: The reviewer's point is well taken, our verbiage was redundant in this context since the problem can be solved without any noise, and so any level of noise would imply at least a degree of error. We will revise it to just "erroneous" in that section.
>
> Q5: The CausalMIL work (citations 32, 33 in the article) has already shown empirically that respecting the MIL assumptions can lead to significant gains in accuracy. It would be hard to adapt our benchmarks to "real world" data because we would need to label the bag-level information for every data point to construct our testing approach. The benefit of synthetic tests like this are:
>
> 1. Once it is shown to fail, it does not matter what any other dataset may look like, because an existence certificate of non-MIL behavior has been achieved.
> 2. This makes it easy to apply our tests to any MIL model (e.g., many computer vision-based MIL algorithms would be hard to adapt to NLP/cyber security problems).
>
> We have added new text to an all-reviewer rebuttal that we believe will satisfy this concern.
>
> > Lines 51-66 discusses the paper’s sections. It would strengthen the paper if instead the paper discusses at least a few of the algorithmic unit tests/MIL assumptions that would give the reader a better understanding of what the paper is trying to do. Reading the introduction does not give specific information like this. Also highlighting how an example popular MIL method fail an important MIL assumption would also make the introduction much more intriguing.
>
> Alg. 1 and 2 test the fundamental property of the MIL hypothesis: that you can not make a positive prediction ($y =1$) based on the absence of an instance $\boldsymbol{x}_i$  from the larger bag $X$. The difference between them is that in Alg. 1, there is only one instance necessary to make a bag label become positive. In Alg. 2, there are two instances that must occur for a positive label. This distinction is important as it sperates two versions of the MIL problem: the Standard MIL (Alg 1) and Threshold MIL (Alg 2).
>
> Alg 3 tests that a Threshold MIL algorithm must be able to count and recognize two different items to detect a positive bag.
>
> As an example of how an algorithm can fail the MIL model, consider the case of cancer detection. The goal is to detect any _instance_ of a cancerous cell, which indicates the whole sample (i.e., _bag_) is infected. However, noise in the data may cause spurious items to correlate with a non-cancerous prediction: mislabeled data, artifacts from the imaging equipment, the tool used for excising the sample, the clinician who collected the samples, and more could correlate with a benign prediction. A non-MIL model is free to learn to use these spurious anti-correlated features. But in deployment, the physician, hospital, and equipment, all change -- and so the non-MIL model's accuracy may drop. Indeed, the failure of medical ML models to generalize has been a noted problem (see all-reviewer citations), and so ensuring MIL assumptions can help avoid these high-risk failures.
>
> We will add all the above to the introduction.
>
> > Also, there is no mention of reproducibility in the whole paper other than the title and related work. I am genuinely curious about how reproducibility is related to the algorithmic unit tests here, or what “reproducibility” means in this context. some clarification from the authors would be important.
>
> Many papers that study reproducibility focus not on generating the same results, but on issues in procedure that would prevent robust conclusions about utility. This includes methodological issues like not tuning baselines [1], or insufficient statistical tests [2]. Our work falls in this latter group of identifying a methodological issue, in not validating the hypothesis constraints.
>
> 1. https://doi.org/10.1145/3298689.3347058
> 2. https://proceedings.mlsys.org/paper_files/paper/2021/hash/0184b0cd3cfb185989f858a1d9f5c1eb-Abstract.html
>
> > The paper is missing a “preliminaries” section, and it is not self-contained. It ought to contain a section describing the problem setup of MIL.
>
> We attempted to interleave this with the introduction. The only detail we might say is missing is that most MIL algs work by learning an instance classifier $h(x)$, and using $\hat{y} = \max_i h(x_i)$ to make a prediction. We will add to the introduction.

---

> > ### Comment · Reviewer_6q8u · 2023-08-11
> >
> > I thank the authors for writing compact answers to my questions and thoughts. The authors answers improved my understanding of the paper, where I misunderstood a few aspects before that are now clarified. I have increased the score from 3 to 6 accordingly.

---

> > ### Comment · Reviewer_6q8u · 2023-08-11
> >
> > I would suggest improving the paper's writing. There are few grammatical mistakes which makes the paper less readable.
> >
> > 1. Line 370-371: The goal is that models are tested to the properties the properties to have.
> >
> > In addition to many more typo/writing issues reviewer 3dcE has mentioned.
> >
> > Furthermore, a discussion of real-world MIL-benchmarks/settings [1, 2], their construction and if their construction respect/do not respect the MIL assumptions here would strengthen the paper and showcase its novelty.
> >
> > [1] mil-benchmarks: Standardized Evaluation of Deep Multiple-Instance Learning Techniques, https://arxiv.org/abs/2105.01443
> >
> > [2] Multiple Instance Learning: A Survey of Problem Characteristics and Applications, https://arxiv.org/abs/1612.03365

---

> > > ### Author Response · Authors · 2023-08-11
> > >
> > > We appreciate the raised score in light of the clarification. Thank you!
> > >
> > > We have fixed all mentioned typos and will carefully review to correct any remaining typos.
> > >
> > > 2105.01443 : This appears to be 2 versions of the Standard MIL formulation, 1 version of the Threshold MIL, 1 appears to be a version of the extended GMIL per Foulds & Frank, and one that might be a Counting-GMIL per Foulds & Frank.
> > >
> > > 1612.03365: This has three synthetic/semi-synthetic tasks (amongst other real datasets). Newsgroups, Letters, and Gaussian, are all standard MIL.
> > >
> > > We will add these related works and a discussion about prior synthetic MIL datasets to the camera ready. We will emphasize that we are not the first to create synthetic data with the goal of testing specific properties and their identifiability by a MIL model. However, to the best of our knowledge, ours is the first work to create adversarial test sets with a MIL solution, and a non-MIL solution, as a way to test that a model restricts itself to a valid hypothesis.
> > >
> > > 1612.03365 is also an excellent catalog of many real-world datasets for MIL, as you note, that highlights the importance of understanding (1) which version of the MIL hypothesis we wish to leverage, and (2) why it is important to respect the (desired) MIL hypothesis. This will also be worked into the discussion.
> > >
> > > We hope this further clarifies any concerns, and we are again appreciative of your time. Please let us know if any further thoughts remain, and we hope you have a great weekend!

---

### Author Rebuttal · Authors · 2023-08-09

We are pleased most reviewers found our paper readable, sound, and technically novel in identifying a previously undocumented issue in the Multiple Instance Learning literature. One shared note of reviewers was that we could more strongly communicate the importance of this to readers outside the standard MIL audience. We will add the below text to the manuscript and hope it satisfies your concerns.


Algorithms that fail, or intentionally forgo, the MIL constraints may appear to obtain better accuracy "in situ" (i.e., the lab environment). But if it is known that the MIL assumption is true, ignoring it creates a significant risk of failure to generalize "in vivo" (i.e., in real production environments). In the clinical context, this is important as many ML algorithms are often proposed with superior in situ performance relative to physicians [1], but fail to maintain that performance when applied to new clinical populations [2,3,4]. In this case, respecting underlying MIL properties eliminates one major axis of bias between situ and vivo settings and higher confidence in potential utility. In the cyber security space, respecting the MIL nature eliminates a class of "good word" style attacks [5,6,7] where inconsequential content is added to evade detection, an attack that has worked on production anti-virus software [8]. These reasons are precisely why MIL has become increasingly popular, and the importance of ensuring the constraints are satisfied.  For these reasons, we would suggest practitioners/researchers begin with CausalMIL and mi-Net as a solid foundation to ensure they are actually satisfying the MIL hypothesis, and thus avoiding excess risk in deployment.

Notably, this creates a dearth of options when more complex MIL hypotheses are required, as CausalMIL and mi-Net succeed by restricting themselves to the Standard MIL assumption. The creation of MIL models that satisfy this, and other more complex hypothesis, are thus an open line of research that would have potentially significant clinical relevance. Similarly, users with more niche MIL needs may desire to more thoroughly test their models respect the constraints critical to their deployment. Our work has demonstrated that many articles have not properly vetted the more basic MIL setting, and so we suspect other more complex MIL problems are equally at risk.

1. Poore GD, Kopylova E, Zhu Q, Carpenter C, Fraraccio S, Wandro S, Kosciolek T, Janssen S, Metcalf J, Song SJ, Kanbar J, Miller-Montgomery S, Heaton R, Mckay R, Patel SP, Swafford AD, Knight R. Microbiome analyses of blood and tissues suggest cancer diagnostic approach. Nature. 2020 Mar;579(7800):567-574. doi: 10.1038/s41586-020-2095-1. Epub 2020 Mar 11. PMID: 32214244; PMCID: PMC7500457.
2. Abraham Gihawi, Yuchen Ge, Jennifer Lu, Daniela Puiu, Amanda Xu, Colin S. Cooper, Daniel S. Brewer, Mihaela Pertea, Steven L. Salzberg bioRxiv 2023.07.28.550993; doi: https://doi.org/10.1101/2023.07.28.550993
3. Varoquaux, G., Cheplygina, V. Machine learning for medical imaging: methodological failures and recommendations for the future. npj Digit. Med. 5, 48 (2022). https://doi.org/10.1038/s41746-022-00592-y
4. Wynants L, Van Calster B, Collins G S, Riley R D, Heinze G, Schuit E et al. Prediction models for diagnosis and prognosis of covid-19: systematic review and critical appraisal BMJ 2020; 369 :m1328 doi:10.1136/bmj.m1328

---

### Decision · Program_Chairs · 2023-09-21

**Decision:**

Accept (poster)

**Comment:**

Reviewers were unanimously supportive of this paper, which provides an interesting analysis of subtle failure cases of multiple instance learning algorithms. The author response significantly improved many reviewers' scores; the authors are encouraged to update the final version based on the this discussion, which will likely benefit future readers.